# Reliability Evaluation for Continuous-Wave Functional Near-Infrared Spectroscopy Systems: Comprehensive Testing from Bench Characterization to Human Test

**DOI:** 10.3390/s24072045

**Published:** 2024-03-22

**Authors:** Chenyang Gao, Jia Xiu, Chong Huang, Kaixue Ma, Ting Li

**Affiliations:** 1Institute of Biomedical Engineering, Chinese Academy of Medical Sciences & Peking Union Medical College, Tianjin 300192, China; gaocy@bme.pumc.edu.cn (C.G.); 22110850035@m.fudan.edu.cn (J.X.); 2Philips North America, Carlsbad, CA 92011, USA; chong.huang@philips.com; 3School of Microelectronics, Tianjin University, Tianjin 300072, China; kxma@ieee.org

**Keywords:** functional near-infrared spectroscopy, optical instrument, optical detection, reliability evaluation

## Abstract

In recent years, biomedical optics technology has developed rapidly. The current widespread use of biomedical optics was made possible by the invention of optical instruments. The advantages of being non-invasive, portable, effective, low cost, and less susceptible to system noise have led to the rapid development of functional near-infrared spectroscopy (fNIRS) technology for hemodynamics detection, especially in the field of functional brain imaging. At the same time, laboratories and companies have developed various fNIRS-based systems. The safety, stability, and efficacy of fNIRS systems are key performance indicators. However, there is still a lack of comprehensive and systematic evaluation methods for fNIRS instruments. This study uses the fNIRS system developed in our laboratory as the test object. The test method established in this study includes system validation and performance testing to comprehensively assess fNIRS systems' reliability. These methods feature low cost and high practicality. Based on this study, existing or newly developed systems can be comprehensively and easily evaluated in the laboratory or workspace.

## 1. Introduction

As people’s standard of living increases, healthcare services require state-of-the-art technologies to improve quality and patient experience. In recent years, bio-optical technology has been applied in a number of important diagnostics and therapy procedures. Functional near-infrared spectroscopy (fNIRS) has been used for many years for non-invasive brain monitoring [1,2,3]. fNIRS utilizes the modified Beer–Lambert law, which calculates chromophore concentrations from optical intensity variations, conducting relative changes in oxy-hemoglobin and deoxy-hemoglobin concentrations [4,5]. Prior studies demonstrate that increased neuronal activity is associated with increased local cerebral blood flow and oxygen delivery in response to the higher metabolic demands of the brain. This process refers to neurovascular coupling that has been frequently reported using fNIRS-based brain imaging [6,7].

As a reliable, non-ionizing, low-cost technology with few environmental constraints, fNIRS has been widely studied [8,9,10,11], and a variety of fNIRS instruments have been developed [12,13,14,15] recently. Intrinsic techniques categorize fNIRS instruments into continuous wave (CW), frequency domain, and time domain [14]. The CW-fNIRS, which is most used, records changes in the amplitude and intensity of the emitted CW light, leading to relative changes in HbO_2_ and Hb concentrations [15,16,17].

As more and more CW-fNIRS instruments are developed and commercialized, various instrument test methods and standards have been reported [16,17,18,19]. However, most of these methods have designed only in vivo experiments adapted to the application scenario of the instrument to determine its performance directly from the experimental results. The literature also incorporates simple testing (e.g., blood model, vascular occlusion experiment, and Valsalva movement experiment [16,17,18,19]) to verify the system's performance in a specific circumstance. Currently, CW-fNIRS technology still lacks a systematic and comprehensive instrumentation evaluation method. When the device does not meet clinical needs due to poor performance or when users need detailed device specifications for a particular application, systematic and comprehensive testing is required to aid in device improvement or selection. In addition, when the CW-fNIRS instrument fails to detect the desired physiological signals, comprehensive, systematic, and rational evaluation methods can help determine and locate problems. With the proposed rational evaluation methods, developers can assess the capability level of an instrument in a certain characteristic and adjust the design ideas in time.

This study used a CW-fNIRS system developed in our lab to test the proposed evaluation methods. The tests include model evaluation and functional brain imaging. The evaluation method of the CW-fNIRS system for functional brain imaging was summarized, demonstrating great reliability. This study will promote the development and assessment of CW-fNIRS instrumentation.

## 2. Methods

Before the fNIRS system can be formally used for brain functional imaging, it is essential to carry out experiments to verify the basic performance of the system, especially in brain functional imaging, thus guaranteeing a complete performance characteristic. The system performance specifications can be described using luminous power, dark noise, stability, interference of background light, and inter-channel interference. By experimenting on some tissue parameters, the validity of the system can be reflected in the measurement process. In addition, some classical cognitive paradigms are helpful in determining the performance of brain functional imaging before clinical use. Figure 1 shows the main evaluation indicator of this study.

### 2.1. Test Object

In this study, our self-developed CW-fNIRS system was taken as the test object. The system architecture was commonly used in the current CW-fNIRS technology (Figure 2a). With the support of an industrial stabilized power supply, a data acquisition card (PXIE-6368, National Instrument, Austin, TX, USA) was used to control the source, detector, and data transmission. The light source and the detector transmit/receive photons through optical fibers in contact with the human scalp. The light sources are laser diodes (LD), and two wavelengths (780 nm and 850 nm) are coupled as a group using a fiber splitter. LD light source, developed by QSI (QL78M8S-A, ProPhotonix, Hertfordshire, UK), adopts sinusoidal modulation of different frequencies, and the digital locking technology of the upper computer can suppress noise in signal transmission to a large extent. The detector is an Avalanche photodiode (APD) developed by Hamamatsu (C12703-01, Hamamatsu Photonics, Hamamatsu, Japan) with an amplifier gain of 1.5 × 10^8^ V/W, a bandwidth of 100 kHz, and an equivalent noise power of 0.02–0.04 pW. The specific device has 16 LDs (8 groups) and 32 detectors. The system sampling rate can be set from 1 Hz to 200 Hz. The appearance of the system is displayed in Figure 2b.

### 2.2. System Performance

The performance of the instrument was evaluated from four aspects: optical safety, stability, robustness, and sensitivity. Luminous power was the most relevant parameter for patient optical safety. For stability, long-term (2 h) monitoring of light intensity output is desired according to typical applications, although the received optical signal is usually relatively weak [20]. Because the heat generated by the light source will affect the measurement performance of the equipment. Prior to starting the experiment, we turned on the LD sources for 20 min to reach a stable temperature. The ability to resist system noises such as XX and YY is critical for robustness. For sensitivity, the real-time output of the monitoring results was observed and compared by changing the content of the scattering material.

#### 2.2.1. Optical Safety

In a fiber-contact scheme where the source spot size is determined, the fluence varies only as the luminous power. The luminous power of each light source was measured with a luminous power meter developed by Thorlabs, Newton, NJ, USA (PM100D, Figure 3a).

#### 2.2.2. Stability

The optical probe contains a black foam pad, which carries five fiber tips. This probe is attached to a self-made solid phantom, which simulates tissue optical properties. The phantom was made of gel, which is a substitute for biological tissue. In the production process, a certain proportion of Indian ink (Sennelier) and white particles were added to simulate the absorption and scattering of biological tissue. More details of the phantom can be found in the reference [21]. The test is set in a dark room to minimize the impact of outside light (shown in Figure 3b). The recorded data must be segmented to analyze the stability, and the drift relative error must be calculated. The calculation formula is as follows:(1)$$Error=\frac{1}{k}\sum_{n=1}^{k}\frac{S_{n}-\frac{1}{2}\left(s_{n-1}+s_{n+1}\right)}{S_{n}}\times 100\%+std$$
where *k* represents the number of segments, *S_n_* denotes the mean of the nth segment of the signal, and std stands for the variance of the entire signal.

#### 2.2.3. Dark Noise

Dark noise includes DC drift and noise fluctuation. DC drift is introduced by the input offset voltage and input bias current of the op-amp and the on-resistance of the analog switch during the actual operation of the circuit. The noise fluctuation is related to the signal detected by the system, namely voltage value, with the scenario of no light source and no influence of external light. The noise voltage of the system can be obtained by subtracting the DC drift of the system.

We evaluated the impact of the background noise caused by environmental light. The voltage value measured within a certain time was *U*_1_. The measurement was repeated with the room light turned on, and the voltage with an identical period was *U*_2_. The impact of background light interference is calculated with the following formula:(2)$$\frac{U_{2}-U_{1}}{U_{1}}\times 100\%$$

#### 2.2.4. Inter-Channel Interference

While the system adopts multi-channel detection, the inter-channel interference between each is inevitable. This experiment measures the level of inter-channel interference of the system.

In the darkroom, we first turned off the light source of the device and covered Channel 4 with black tape. When the detected light intensity curve stabilized, we turned on the light source and recorded the signal through an uncovered Channel 1 with a modulated rectangular wave. More specifically, the crucial operation in measuring inter-channel interference involves covering one channel while leaving the other uncovered to record the periodic changes in the light signal. Channel 1 and Channel 4 do not have specific positions or channel requirements. Researchers can freely choose any two channels for inter-channel interference testing during the replication process.

#### 2.2.5. Blood Model

The blood model experiment detects the instrument's validity by measuring the optical density variations produced by the simulated in vitro model during the dripping of blood [19].

Intralipid solution is a highly scattering medium with very weak absorption of near-infrared light. It is commonly used to simulate the scattering properties of biological tissues. Thus, the changes in blood oxygen concentration and local blood volume in the target tissue can be reflected via the measurement in vitro.

A cylindrical container containing a thick polyethylene bag contained 375 mL of phosphate-buffered saline (PBS) and 10 mL of 10% intralipid solution. The container was placed on the magnetic stirrer (shown in Figure 3c). During the experiment, the light density curve was observed in real-time through the display. After a 2 min baseline recording, 10 mL of blood was added every 1 min to the solution until the curve no longer changed. The final injection volume could be considered as the system's sensitivity (also can be seen as the dynamic range).

### 2.3. Validity in In Vivo Hemodynamic Test

fNIRS devices detect functional activities of biological tissue by measuring hemodynamic parameters. Altering blood circulation or blocking blood vessels can produce changes in hemodynamic parameters, which can be used to evaluate the in vivo measurement capability of fNIRS instruments.

The in vivo hemodynamic test includes the vascular occlusion test and the Valsalva movement test. The selection and operation sequence of these two experiments and the N-back experiment introduced in Section 2.4.1 follow the step-by-step principle. From the forearm to the brain, from simply measuring changes in blood oxygen signal when the vessels were blocked to changes in blood oxygen stimulated by specific tasks. The blood oxygen response changes from strong to weak, and the signal detection changes from easy to difficult. Thus, the system performance can be verified layer by layer, helping developers to characterize the instrument's blood oxygen signal detection ability.

#### 2.3.1. Vascular Occlusion Experiment

A mercury sphygmomanometer vascular occlusion system was applied to the upper arm for blood pressure measurement, and the light source and detector were fixed in the middle part of the forearm with a sponge and elastic band. During the experiment, the vascular occlusion pressure was directly increased to 220 mmHg to achieve the complete blockade of static and arterial blood flow.

Before the experiment began, subjects were instructed to relax and rest (the participants’ participation in the experiment is shown in Figure 4a), and then the instrument was turned on and baseline data recorded. After baseline recording was completed, the inflatable vascular occlusion cuff was rapidly pressurized with a pressurized balloon. When the set value is reached, the pressure-holding phase is initiated. Finally, the inflatable vascular occlusion was quickly deflated with a pressurized balloon. During this period, the forearm hemodynamic changes were recorded using a near-infrared instrument throughout the procedure.

#### 2.3.2. Valsalva Movement Experiment

Valsalva movement instructed the subject to perform strong closed breathing movement, deep inhale, close the glottis, and then force to exhale against the closed epiglottis. The chest pressure increased at this stage and produced a blocking effect similar to the vascular occlusion experiment in the concentration of hemoglobin in the brain during the movement.

The subjects sat in a comfortable chair, kept their body relaxed, and performed a series of operations according to the voice cues (the participants’ participation in the experiment is shown in Figure 4b). The subjects remained relaxed from the beginning to 25 s. At 25 s, there was a sound stimulus (start), and the subject performed the Valsalva movement and kept it for 30 s. After 30 s, a second acoustic stimulus (end) appeared, at which time the subject began to calm down and continued for some time.

### 2.4. Validity in Brain Functional Imaging

N-back is one of the classic fNIRS experimental paradigms for brain function detection, and its blood oxygen activation signals and regions have been well studied, so we selected it for the brain functional imaging test.

#### 2.4.1. Participants and Paradigm

Seventeen healthy volunteers (nine women) took part in this study. The age of the participants was 19 to 26 (M = 22.7; SD = 1.87). They were all right-handed, had no history of neurological or psychiatric diseases, and had normal hearing. Their attention was concentrated, and they had no signs of hyperactivity disorder.

The experiment was conducted in a dark and quiet room to avoid any interference from the environment. All participants were seated in a comfortable chair in a room with good air condition to avoid the influence of environmental stress and were asked to minimize their head movements, reduce swallowing movements, and avoid swings of mood throughout the entire experiment (the participants’ participation in the experiment is shown in Figure 5c).

The experimental paradigm consisted of two tasks: 2-back and 0-back. In both tasks, a letter appeared on a screen at the same time, and the subjects were asked to make a judgment. In the 2-back task, subjects needed to judge whether the current letter was the same as the letter that appeared next to it. If the letter was the same, they were asked to press the left mouse button; if not, they pressed the right mouse button. In the 0-back task, the subjects only needed to judge whether the current letter was “A”; if it was, they should press the left mouse button; otherwise, they should press the right mouse button. The experimental process is shown in Figure 5a. There were six blocks in the experiment; each block lasted 90 s, including 30 trials. The 2-back task and 0-back tasks were shown alternately. In each task block, ten letters were set as yes cases. The rest period lasted 60 s between the blocks. In the task, subjects were presented with a sequence of 30 letters. Each letter was presented for 500 ms, and there was a 2.5 s interval during which the screen was blank. Before the experiment began, the task requirements were explained to the subjects, and then the task exercises were performed to ensure that the subjects could achieve more than 70% accuracy in the exercise task.

#### 2.4.2. Data Acquisition

The fNIRS signals were acquired at a sampling frequency of 100 Hz. A total of 20 channels covering the frontal and the parietal lobe, each of the left and right sides of the brain had 10 detection channels with a spacing of 3 cm. Figure 5b shows the position arrangement of the fNIRS probe channel detection system.

#### 2.4.3. fNIRS Signal Analysis

fNIRS data were analyzed offline in MATLAB R2021a (MathWorks, Natick, MA, USA). The program separated the photoelectric marking signal, which was used to calibrate the starting point of the trial. The optical signals from the fNIRS acquisition device were converted to oxygenated (HbO) and deoxygenated (HbR) concentration changes using the modified Beer–Lambert law. The differential path length factor (DPF) was set as DPF785 = 6.0 and DPF850 = 5.2. Then, the data were segmented, the baseline was corrected, and the data were stitched according to the experimental design.

## 3. Results

### 3.1. System Performance

#### 3.1.1. Luminous Power

The maximum light power of the system light source was less than 50 mW (Table 1). According to the ANSI Z136 series standards [22], it is classified as a Class 3 device; its advantages are that it is relatively nondestructive, harmless, has no ionizing radiation, is low dose, and will not cause damage to human tissue cells.

#### 3.1.2. Stability

The system stability was measured for one hour under the specified conditions. The whole experimental process was divided into six modules, which lasted for 2 h. The relative error of drift was calculated according to the formula (Figure 6). The drift error of the system is less than 0.002%, which has little influence on the actual measurement and can be ignored.

#### 3.1.3. Dark Noise

The experimental results of dark noise (Table 2) demonstrated the ability of the system to operate stably for a long time. In the recording process, the mean and standard deviation of the dark noise was maintained at the level of 10^−6^ V. Compared with the signal amplitude level of 10^−2^ V, the device achieves a signal-to-noise ratio of 80 dB, meeting the requirements of most signal acquisition scenarios.

#### 3.1.4. Interference of Background Light

In the experiment, U1 and U2 were measured for 8 min. The maximum background light interferences of all channels are shown in Figure 7. The background light interference degree could be calculated as less than 1%.

#### 3.1.5. Inter-Channel Interference

Inter-channel interference (Figure 8) showed that turning on and off the light source (Channel 1) has an influence on the adjacent channel (Channel 4), but the influence range is 10^−4^ V, which is neglectable compared with the measurement signal of 10^−1^ V.

#### 3.1.6. Blood Model Testing

Figure 9 shows the real-time recording of the optical density value during the experiment. As the experiment continued to add samples, the optical density curve had an obvious upward trend. This phenomenon continued until 0.1 mL was added. At this time, the curve change was not obvious. Therefore, in terms of sensitivity, this system can detect at least 0.2% changes in hemodynamic concentrations.

### 3.2. Validity in In Vivo Hemodynamic Test

#### 3.2.1. Vascular Occlusion Experiment

The experimental results are shown in Figure 10a. In the 30 s of the experiment, blood flow was constrained by inflating and pressurizing the sleeve bag. As the tissue cells continued to consume oxygen, the content of oxygenated hemoglobin and deoxygenated hemoglobin in the measured blood gradually decreased. When the vascular occlusion is deflated and decompressed at 90 s, the influx of fresh blood causes the two hemoglobin levels to resume to baseline gradually.

#### 3.2.2. Valsalva Movement Experiment

The time curves of hemoglobin changes during the Valsalva movement are shown in Figure 10b. In the initial stage, at 25 s, a large amount of oxygenated hemoglobin flows to the brain. Afterward, the oxygenated hemoglobin began to shift to deoxygenated hemoglobin. Therefore, the concentration of deoxygenated hemoglobin increased while the concentration of oxygenated hemoglobin decreased. When the subject stopped the Valsalva movement and began to remain relaxed at 45 s, the oxygenated and deoxygenated hemoglobin levels began to resume to baseline.

### 3.3. Validity in Brain Functional Imaging

During the working memory task, subjects showed significant oxygen activation in the 2-back ventrolateral prefrontal cortex (Figure 11), which was previously implicated in storing and processing verbal information.

Gender differences in brain function varied with task stages (Figure 11), consistent with prior studies. The 90 s block was divided into two 45 s data segments. The fNIRS brain activation in the two segments was analyzed, respectively. The results showed that the sex difference of HbO_2_ mainly existed in the anterior segment. The HbO_2_ concentration of the prefrontal cortex (PFC) in male subjects increased quickly at the beginning of the task, exceeded that of female subjects, and began to decline about one-third of the way through the task, then approached that of female subjects in the second half. This phenomenon indicated that males need more time to adapt to this task, so they need to call more brain resources and have stronger brain activation in the early stage, and the brain activation level declined after the subjects began to adapt to this task. In women, the brain activation curve remained flat throughout the tenure. It also suggested that women are more suitable for such experiments because of their efficient adaptability. The possible reason from the perspective of heredity and natural evolution is that females usually use symbols to mark information, while males are good at using spatial location to mark information [23].

The results of blood oxygen activation showed that the experimental design was feasible and the measurement of experimental data was as expected, which proved that the device could be applied to detect brain function [24].

## 4. Discussion and Conclusions

The testing results proved that the instrument has a certain degree of safety based on the luminous power of light sources, could resist environmental and inter-channel interference, and ensured stability under long-time work conditions. It also performed well in isolated blood models, including blood flow measurement in the forearm and brain and detection of hemodynamic alternations during the cuff experiment. Moreover, the system can achieve the expected results in the classical n-back paradigm. In summary, the system can be reliably applied to various experimental scenarios of brain function detection.

We have gradually deepened the testing of the system, and if mistakes were made in some steps, they could be quickly corrected. Firstly, to ensure the safety requirements, we need to evaluate its basic capabilities, including stability and robustness. Then, simple model experiments were performed, from the forearm to the brain, ranging from simple measurements of changes in blood oxygen signals during vascular blockage to changes in blood oxygen under task-specific stimuli. After these preparations, the usual experimental tasks were finalized, and the results were tested after the data had been processed to ensure that the device could be used for functional brain imaging. Using this protocol, developers and researchers can clearly identify where and how the CW-fNIRS system problems emerged and systematically assess the instrument's performance as a third-party test facility or self-assessment.

All the experiments involved in this study were easy to carry out, and the equipment required for the experiments were commonly used materials in optical laboratories. The experiments are simple to perform, and the overall time does not exceed 2 h, except for system stability. Finally, if it is not convenient to perform N-back experiments, other classical paradigms can be used instead. In general, these methods are highly operational and low-cost. In addition, some general test methods for system performance can be easily transferred to other optical inspection instruments, such as other DOT instruments, laser speckle imaging, optical coherence tomography, and photo thoracoscopy instruments.

In this study, we used our self-developed fNIRS instrument instead of a commercial instrument. This instrument has been shown to perform well by successfully detecting brain activity in a motor imagery task. Therefore, we could test the instrument in more fundamental aspects, such as inter-channel interference and interference from background light. Our study provides reference and guidance value for evaluating self-developed devices. In order to fully promote the clinical application of biomedical optics, we suggest that more DOT evaluation systems be established with reference to our study.

However, the above methods still have some limitations. Firstly, the proposed method was only applicable to CW-NIRS equipment, which has some limitations in its application. Secondly, since the testing device is applied to the human body, safety should be the first factor to be considered, but there are no more safety assessment indicators. Thirdly, we did not use short channels in the N-back experiment for the cerebral level test. And that may be the reason why we do not measure hemodynamic activity during 0-back. Short channels combined with an adaptive filtering algorithm can effectively remove global physiological noise and surface interference [25]. fNIRS’s unique advantages and wide application scenarios encourage us to continuously add new indicators, improve indicators and standards, and expand the testing scope in future work.

We sincerely hope that the protocol and results of this study will be of assistance to research groups who intend to design new systems or evaluate existing ones and provide them with the basic insurance needed in fNIRS research.

## Figures and Tables

**Figure 1 sensors-24-02045-f001:**
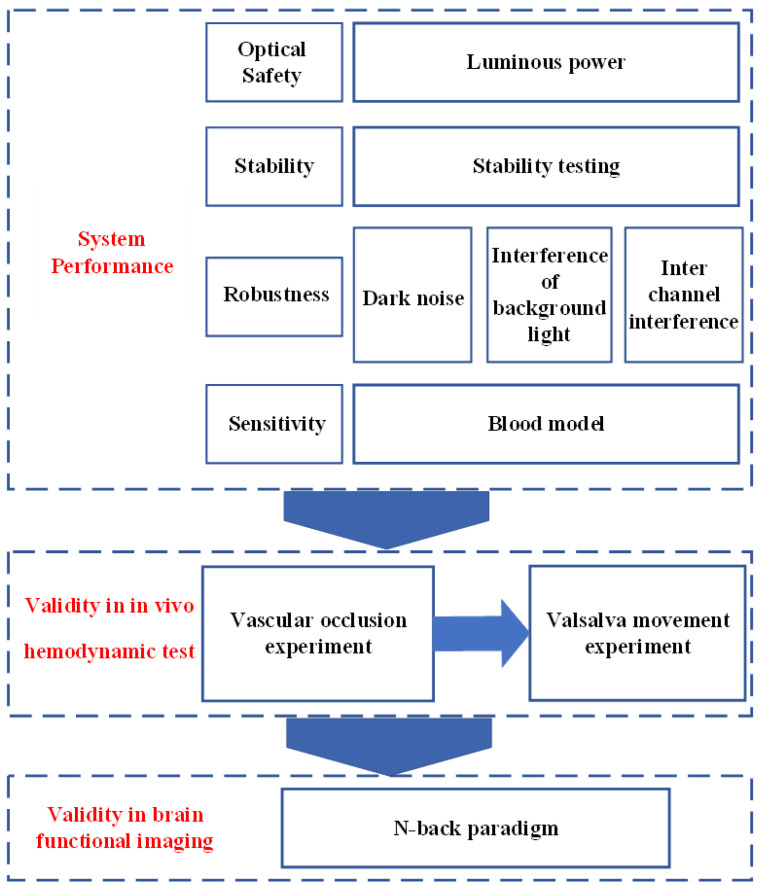
Evaluation methods logic diagram.

**Figure 2 sensors-24-02045-f002:**
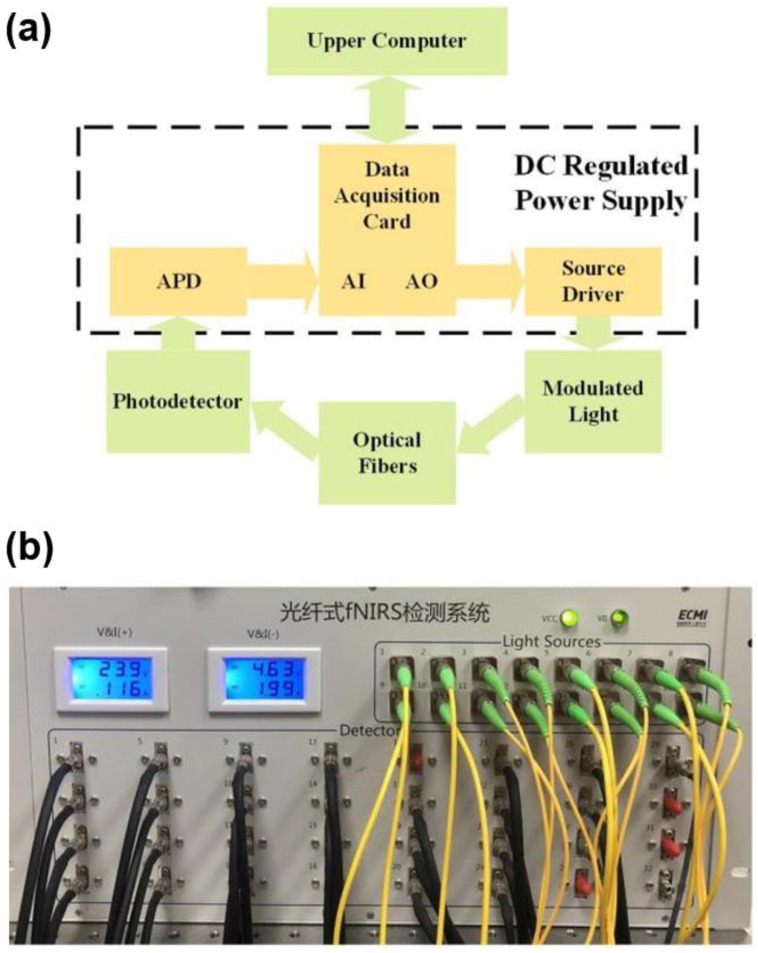
(**a**) Hardware block diagram, and (**b**) the appearance of the system.

**Figure 3 sensors-24-02045-f003:**
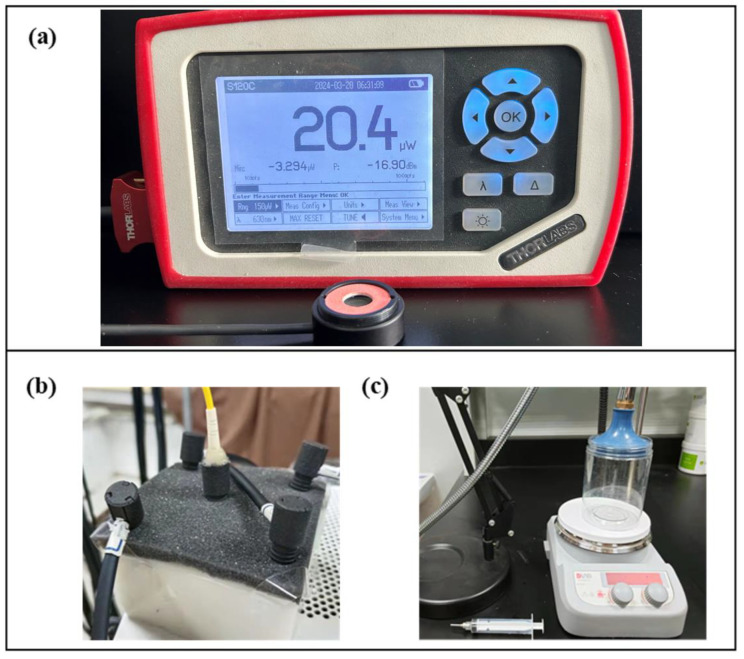
Experimental auxiliary equipment diagram. (**a**) The appearance of the luminous power meter. (**b**) A black sponge attaches a flexible probe to the optical phantom for evaluating stability, interference of background light, and inter-channel interference. (**c**) Blood model measurement system.

**Figure 4 sensors-24-02045-f004:**
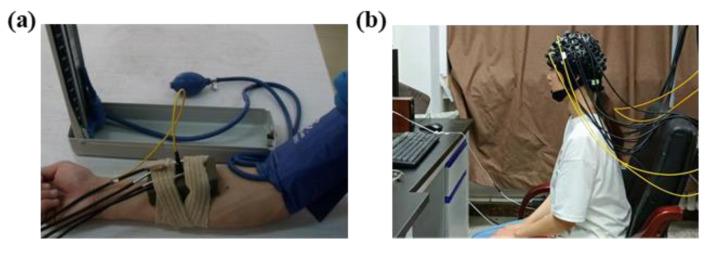
(**a**) Illustration of the vascular occlusion experiment. (**b**) Illustration of the Valsalva movement experiment.

**Figure 5 sensors-24-02045-f005:**
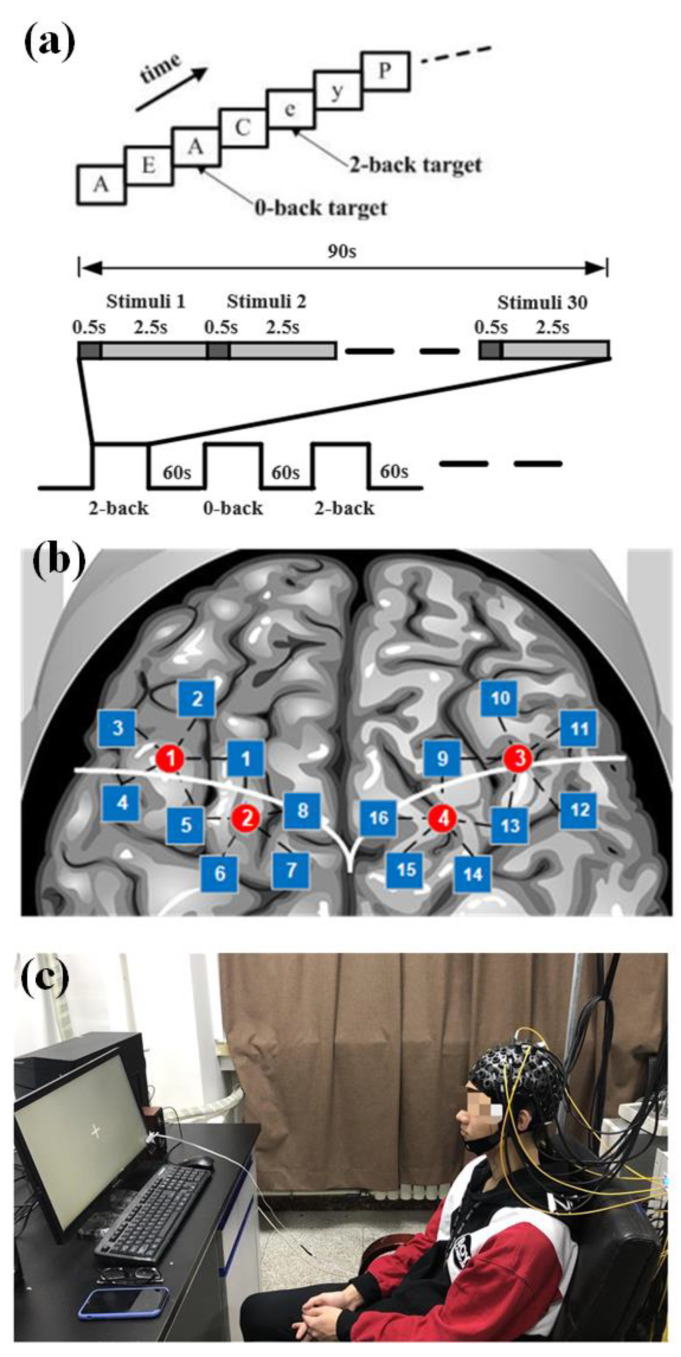
(**a**) Schematic diagram of experimental paradigm. (**b**) Left and right fNIRS channel number and brain region distribution. The red circle represents the light source, while the blue square represents the detector, and the number is their channel number. The white line is the boundary between the frontal and parietal lobes. (**c**) Participants participating in the experiment.

**Figure 6 sensors-24-02045-f006:**
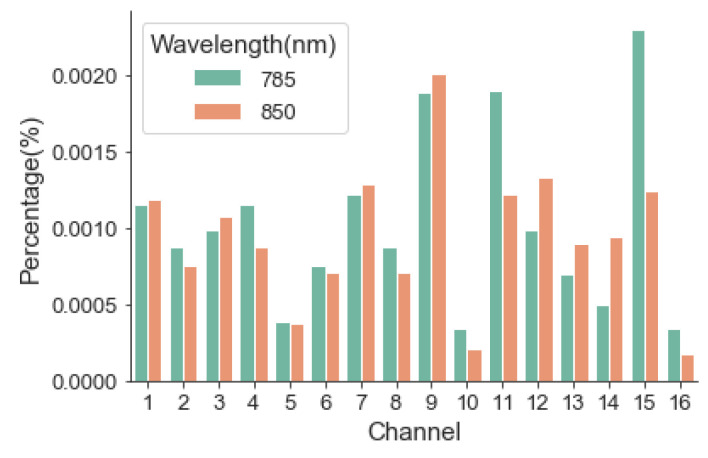
Relative excursion error of each channel and each wavelength.

**Figure 7 sensors-24-02045-f007:**
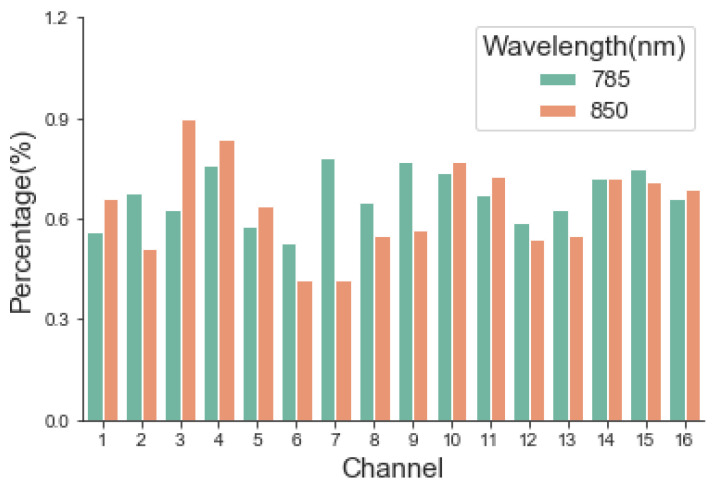
Maximum background light interference of each channel.

**Figure 8 sensors-24-02045-f008:**
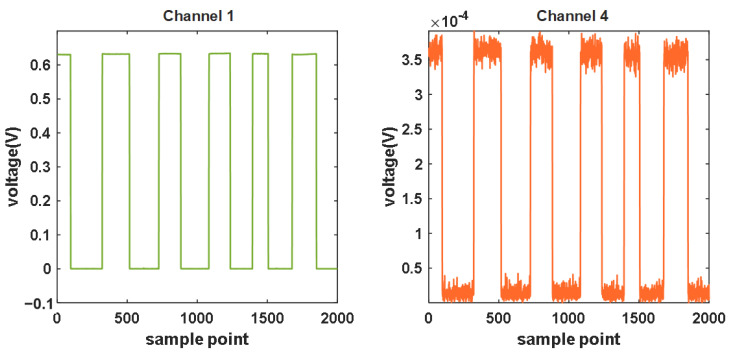
The detected voltage curves of Channel 1 and Channel 4.

**Figure 9 sensors-24-02045-f009:**
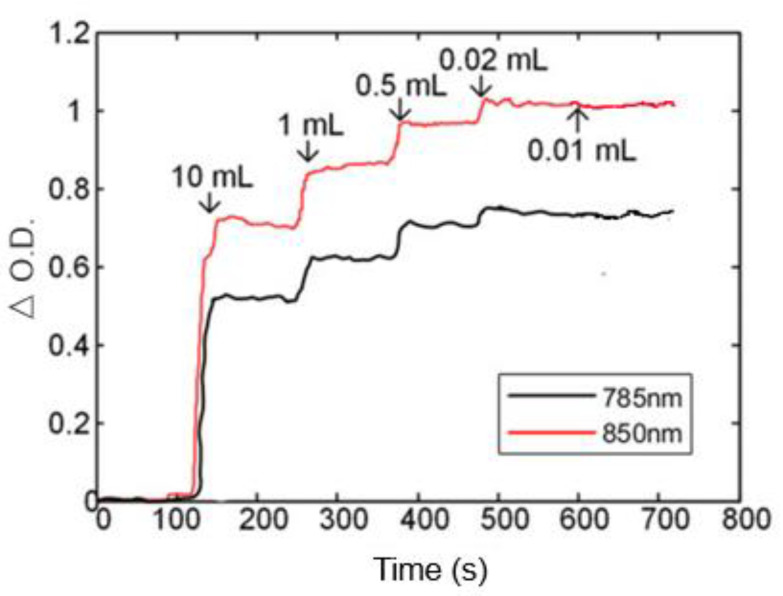
Blood model experiment results.

**Figure 10 sensors-24-02045-f010:**
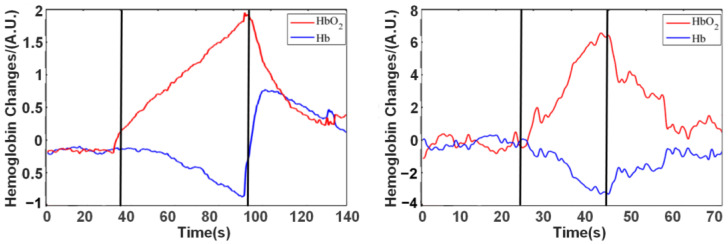
(**a**) Results of vascular occlusion experiment. A.U. (arbitrary unit) represents the relative strength. (**b**) Results of Valsalva movement experiment. A.U. (arbitrary unit) represents the relative strength. The black vertical lines represent the beginning and end of the task.

**Figure 11 sensors-24-02045-f011:**
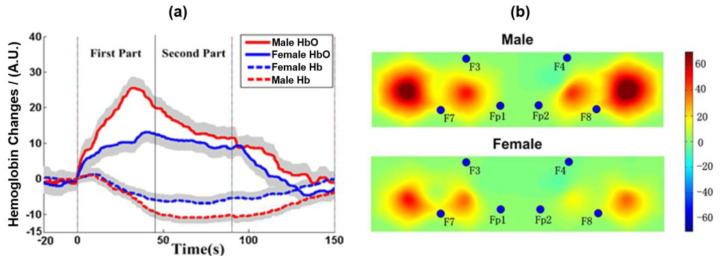
(**a**) The average value of HbO_2_ and Hb during the experiment. The gray area indicates a standard error (SE). (**b**) The average activity of HbO_2_ during the 2-back task.

**Table 1 sensors-24-02045-t001:** The luminous power of light sources.

Light Source	1	2	3	4	5	6	7	8
785 nm (mW)	30.5	30.6	38.4	29.1	43.4	36.8	29.7	39.6
850 nm (mW)	33.8	31.3	45	33.7	31.3	49	38.3	31.5

**Table 2 sensors-24-02045-t002:** Means and standard deviations of dark noise.

	Mean Values (10^−6^ V)		Standard Deviations (10^−6^ V)	
Channel	785 nm	850 nm	785 nm	850 nm
Channel 1	5.58	6.06	6.26	6.79
Channel 2	5.11	5.16	5.76	5.82
Channel 3	4.74	4.77	5.34	5.37
Channel 4	4.84	4.84	5.46	5.46
Channel 5	4.44	4.46	5.02	5.03
Channel 6	4.68	4.42	5.28	4.99
Channel 7	4.44	4.20	5.02	4.74
Channel 8	4.54	4.55	5.12	5.13
Channel 9	4.35	4.40	4.91	4.97
Channel 10	5.01	4.77	5.64	5.40
Channel 11	4.51	4.33	5.10	4.88
Channel 12	4.51	4.50	5.09	5.08
Channel 13	5.73	5.45	6.43	6.11
Channel 14	4.53	4.32	5.12	4.87
Channel 15	4.63	4.71	5.22	5.31
Channel 16	4.29	4.39	4.84	4.95

## Data Availability

Due to the research team's data management policy, data underlying the results presented in this paper are not publicly available at this time but may be obtained from the authors upon reasonable request.

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
