# Peer review of "Reliability Evaluation for Continuous-Wave Functional Near-Infrared Spectroscopy Systems: Comprehensive Testing from Bench Characterization to Human Test"

_sensors, 2024, doi:10.3390/s24072045_

Round 1
Reviewer 1 Report
Comments and Suggestions for Authors
The paper concerns the evaluation of the reliability for a continuous-wave functional near-infrared spectroscopy (CW-fNIRS) system by several benchmark tests. The authors have chosen the power of the light source (optical safety), the stability of the data, the amplitude of the dark noise, the background signal, the inter-channel interference signal, and the sensitivity, as the benchmarks. The reliability of the system has been confirmed by the in vivo hemodynamic test. The authors show excellent benchmark results for their laboratory-built CW-fNIRS system and demonstrate that it is capable of monitoring the brain function with a high sensitivity.
While the paper clearly describes the high performance of the authors’ system, it has serious problems as a research article on the establishment of a reliability evaluation method of a CW-fNIRS system, as listed below. Because of these problems, the reviewer does not recommend the paper for publication in Sensors.
1. The authors do not show any criterion for judging the reliability from the benchmark tests, such as the upper limit of the laser power for a measurement in safe, the detection limit of ΔOD necessary for detecting a brain function. The authors just qualitatively describe the results with phrases such as “will not cause damage”, “the noise would not affect”.
2. The evaluation method is not reproducible by readers because the experimental setups are not comprehensively written in the main text or shown in Figure 3. For example, the reviewer does not understand what Channels 1 and 4 in the section 2.2.4 (page 5) are or which they are in Figure 3b (page 4). The reviewer does not understand what “a certain proportion” and “white particles” mean, respectively, in the section 2.2.2 (page 4).
3. The explanation on the variables in the equation (1) is missing (page 5). Also, the drift is unable to be evaluated with the equation (1), because the authors subtract an average of Sn-1 and Sn+1 from Sn. This subtraction fully cancels the effect of a drift that occurs linearly with time.
Author Response
We are grateful for the Editor's comments and appreciate the reviewers’ constructive comments. The comments have helped us to make the paper completer and more understandable. Below are our responses to the reviewers’ comments. All revisions in the revised manuscript are highlighted by yellow color.
The paper concerns the evaluation of the reliability for a continuous-wave functional near-infrared spectroscopy (CW-fNIRS) system by several benchmark tests. The authors have chosen the power of the light source (optical safety), the stability of the data, the amplitude of the dark noise, the background signal, the inter-channel interference signal, and the sensitivity, as the benchmarks. The reliability of the system has been confirmed by the in vivo hemodynamic test. The authors show excellent benchmark results for their laboratory-built CW-fNIRS system and demonstrate that it is capable of monitoring the brain function with a high sensitivity.
While the paper clearly describes the high performance of the authors’ system, it has serious problems as a research article on the establishment of a reliability evaluation method of a CW-fNIRS system, as listed below. Because of these problems, the reviewer does not recommend the paper for publication in Sensors.
- The authors do not show any criterion for judging the reliability from the benchmark tests, such as the upper limit of the laser power for a measurement in safe, the detection limit of ΔOD necessary for detecting a brain function. The authors just qualitatively describe the results with phrases such as “will not cause damage”, “the noise would not affect”.
Response: For the safety indicators of laser power, we supplemented the corresponding reference standards (ANSI Z136 series standards). However, for the detection limit of ΔOD necessary for detecting a brain function and the upper limit of dark noise, we did not find corresponding standards/regulations. Instead, we modified the conclusive statements in the manuscript to descriptive statements. For example, "noise would not affect" was changed to "the device achieves a signal-to-noise ratio of 80 dB, meeting the requirements of most signal acquisition scenarios."
- The evaluation method is not reproducible by readers because the experimental setups are not comprehensively written in the main text or shown in Figure 3. For example, the reviewer does not understand what Channels 1 and 4 in the section 2.2.4 (page 5) are or which they are in Figure 3b (page 4). The reviewer does not understand what “a certain proportion” and “white particles” mean, respectively, in the section 2.2.2 (page 4).
Response: Thank you very much for your valuable feedback. In this study, we do not consider selecting data/equipment to achieve better test results when choosing the testing equipment/channels. Therefore, in the replication process, all the recorded equipment mentioned in this study can be flexibly adjusted according to the laboratory's own situation. We believe that if a method does not depend on the equipment used, it demonstrates its general feasibility. We also look forward to other researchers being able to obtain the same results using different recording equipment, or, if the expected results are not obtained, we are also willing to discuss and exchange ideas with them. More specifically, the crucial operation in measuring inter-channel interference involves covering one channel while leaving the other uncovered to record the periodic changes in light signal. Channel 1 and Channel 4 do not have specific positions or channel requirements. During the replication process, researchers can freely choose any two channels for inter-channel interference testing. The naming of the channels here is merely to emphasize the requirement for two independent channels. As for the stability section, "a certain proportion of Indian ink (such as Sennelier) and white particles" is the method used in our laboratory to make the phantom. More details of the phantom can be found in the reference [1]. During the replication process, researchers can rely on laboratory experience to prepare phantoms with properties similar to biological tissues, which will not affect the reproducibility of the results.
[1] Boas, D. A. Diffuse photon probes of structural and dynamical properties of turbid media: theory and biomedical applications. PhD thesis. University of Pennsylvania 1996.
- The explanation on the variables in the equation (1) is missing (page 5). Also, the drift is unable to be evaluated with the equation (1), because the authors subtract an average of Sn-1and Sn+1from Sn. This subtraction fully cancels the effect of a drift that occurs linearly with time.
Response: Thank you very much for your correction. The explanations of the variables have been added below the formula. As for the formula itself, the submission of the initial draft omitted the part regarding "std," and we have made the necessary correction to the formula.
We believe that these revisions have strengthened the quality of the manuscript and better align it with the requirements of the journal. We will continue to refine the manuscript based on your suggestions and welcome any further feedback you may have on the revised version.
Once again, we sincerely appreciate your review and guidance. We look forward to hearing from you.

Reviewer 2 Report
Comments and Suggestions for Authors
This is a well orgnized paper on comprehensive evaluation for continuous-wave functional near-infrared spectroscopy systems.
There are syntax errors, such as "with an amplifier gain of 1.5×108 V/W...." in line 88, "level of 10-6 V" in line 268, and so on.
Comments on the Quality of English LanguageFine.
Author Response
We are grateful for the Editor's comments and appreciate the reviewers’ constructive comments. The comments have helped us to make the paper completer and more understandable. Below are our responses to the reviewers’ comments. All revisions in the revised manuscript are highlighted by yellow color.
There are syntax errors, such as "with an amplifier gain of 1.5×108 V/W...." in line 88, "level of 10-6 V" in line 268, and so on.
Response: We corrected the syntax errors in the revised manuscript.

Reviewer 3 Report
Comments and Suggestions for Authors
I read with great interest the manuscript entitled ‘Reliability evaluation for continuous-wave functional near-infrared spectroscopy systems: comprehensive testing from bench characterization to human test’. The document is well-written and comprehensible, and the methodology used is appropriate. However, I think it's important that the authors correct a few points to improve their work, which is already of good quality.
- At first, Figures 2 and 5 are not readable.
- 2.4.1 (Page 6) : The authors used the n-back task and provided information about it in their paper. Still, it would be important for the authors to complete the details of the task to enable readers or other researchers to replicate their research. For example, the number of stimuli presented, the number of targets, the inter-stimulus interval.
- Table 2 (Page 8) : The first column of the table is not in English.
- Lines 318-321 : In this section, the authors pointed out that the hemodynamic activity of male participants increases rapidly at the start of the task and exceeds that of female participants. This activity then decreases and approaches the hemodynamic activity of female participants. But figure 11a shows exactly the opposite.
- Lines 322-329 : Interpretation of the hemodynamic response should be linked to the participants' performance during the n-back task. It therefore seems important to me that the authors take behavioral performance into account in their interpretation. In addition, they should specify the behavioral results in supplementary material.
- Line 336 : Authors must replace 'Conclusion & Discussion' with 'Discussion & Conclusion’
Finally, I have two general questions
- Did the authors use short channels when measuring hemodynamic activity?
- Was hemodynamic activity measured during 0-back? If so, did the authors subtract 0-back brain activity from 2-back brain activity in order to have only the activity related to the executive process?
Author Response
We are grateful for the Editor's comments and appreciate the reviewers’ constructive comments. The comments have helped us to make the paper completer and more understandable. Below are our responses to the reviewers’ comments. All revisions in the revised manuscript are highlighted by yellow color.
I read with great interest the manuscript entitled ‘Reliability evaluation for continuous-wave functional near-infrared spectroscopy systems: comprehensive testing from bench characterization to human test’. The document is well-written and comprehensible, and the methodology used is appropriate. However, I think it's important that the authors correct a few points to improve their work, which is already of good quality.
- At first, Figures 2 and 5 are not readable.
Response: We redraw figures 2 and 5 in the revised manuscript.
- 2.4.1 (Page 6) : The authors used the n-back task and provided information about it in their paper. Still, it would be important for the authors to complete the details of the task to enable readers or other researchers to replicate their research. For example, the number of stimuli presented, the number of targets, the inter-stimulus interval.
Response: There were six blocks in the experiment, each block lasted 90 s, including 30 trails. The 2-back task and 0-back tasks were shown alternately. In each task block, ten letters were set as yes cases. The rest period lasted 60 s between the blocks. In the task, subjects were presented with a sequence of 30 letters. Each letter was presented for 500 ms and there was a 2.5-s interval during which the screen was blank. We completed the details of the task in 2.4.1 of the revised manuscript.
- Table 2 (Page 8) : The first column of the table is not in English.
Response: We corrected the column into English.
- Lines 318-321 : In this section, the authors pointed out that the hemodynamic activity of male participants increases rapidly at the start of the task and exceeds that of female participants. This activity then decreases and approaches the hemodynamic activity of female participants. But figure 11a shows exactly the opposite.
Response: The legend in figure 11a was wrong. We corrected Figure 11a in the revised manuscript.
- Lines 322-329 : Interpretation of the hemodynamic response should be linked to the participants' performance during the n-back task. It therefore seems important to me that the authors take behavioral performance into account in their interpretation. In addition, they should specify the behavioral results in supplementary material.
Response: The accuracy of all subjects in the 2-back task was lower and the response time was longer than the 0-back task. There was no significant gender difference in behavior. We added the behavioral results in supplementary material (Table 1).
Table 1. Behavior results in N-back tasks.
|
|
Female 0-back |
Female 2-back |
Male 0-back |
Male 2-back |
|
Mean response time(ms) |
627
95.3 |
932
84.5 |
772 |
910 |
|
Mean accuracy(%) |
93.7 |
86.2 |
- Line 336 : Authors must replace 'Conclusion & Discussion' with 'Discussion & Conclusion’
Response: We replaced ‘Conclusion & Discussion' with 'Discussion & Conclusion’ in the revised manuscript.
Finally, I have two general questions
- Did the authors use short channels when measuring hemodynamic activity?
Response: We did not use short channels in this study.
- Was hemodynamic activity measured during 0-back? If so, did the authors subtract 0-back brain activity from 2-back brain activity in order to have only the activity related to the executive process?
Response: Hemodynamic activity was not measured during 0-back.
Round 2
Reviewer 1 Report
Comments and Suggestions for Authors
The authors have addressed the issues which the reviewer pointed out. Because the revised manuscript clearly describes the establishment of a reliability evaluation method of a CW-fNIRS system, the reviewer recommends the paper for publication in the present form.
Author Response
Thank you very much.
Reviewer 3 Report
Comments and Suggestions for Authors
I've read the various responses from the authors and I thank them. But I have one last minor comment. The authors specified in their answers that they did not use short channels, nor did they measure hemodynamic activity during 0-back. In my opinion, it would be a good idea for the authors to underline these points in the limits of their work, especially for measurements at the cerebral level.
Author Response
I've read the various responses from the authors and I thank them. But I have one last minor comment. The authors specified in their answers that they did not use short channels, nor did they measure hemodynamic activity during 0-back. In my opinion, it would be a good idea for the authors to underline these points in the limits of their work, especially for measurements at the cerebral level.
Response: Thank you for the suggestion. We added these points in the limits of our work in the Discussion & Conclusion section of the revised manuscript. One reference about short channels was also added accordingly.